# Mechanical Properties, Fractal Dimension, and Texture Analysis of Selected 3D-Printed Resins Used in Dentistry That Underwent the Compression Test

**DOI:** 10.3390/polym15071772

**Published:** 2023-04-02

**Authors:** Anna Paradowska-Stolarz, Mieszko Wieckiewicz, Marcin Kozakiewicz, Kamil Jurczyszyn

**Affiliations:** 1Division of Dentofacial Anomalies, Department of Maxillofacial Orthopedics and Orthodontics, Wroclaw Medical University, Krakowska 26, 50-425 Wroclaw, Poland; 2Department of Experimental Dentistry, Wroclaw Medical University, 50-425 Wroclaw, Poland; 3Department of Maxillofacial Surgery, Medical University of Lodz, 90-647 Lodz, Poland; 4Department of Dental Surgery, Wroclaw Medical University, 50-425 Wroclaw, Poland

**Keywords:** fractal dimension, texture, 3D print, resin, compression test, biomaterial, bone index, dentistry

## Abstract

Three-dimensional printing is finding increasing applications in today’s world. Due to the accuracy and the possibility of rapid production, the CAD/CAM (computer-aided design/computer-aided manufacturing) technology has become the most desired approach in the preparation of elements, especially in medicine and dentistry. This study aimed to compare the biomechanical properties, fractal dimension (FD), and texture of three selected materials used for 3D printing in dentistry. Three biomaterials used in 3D printing were evaluated. The materials were subjected to the compression test. Then, their mechanical features, FD, and texture were analyzed. All the tested materials showed different values for the studied properties. The only statistically insignificant difference was observed for the force used in the compression test. All three materials showed differences in width and height measurements. The difference in the decrease between the compression plates was also significant. For Dental LT Clear, the mean value was 0.098 mm (SD = 0.010), while for BioMed Amber it was 0.059 mm (SD = 0.019), and for IBT it was 0.356 mm (SD = 0.015). The nominal strain also differed between the materials. IBT had the highest mean value (7.98), while BioMed Amber had the smallest (1.31). FD analysis revealed that Dental LT Clear did not show differences in the structure of the material. The other two materials showed significant changes after the compression test. Texture analysis (TA) revealed similar results: BioMed Amber resin showed significantly less pronounced texture changes compared to the other two materials. BioMed Amber also showed the most stable mechanical properties, whereas those of IBT changed the most. Fractal analysis revealed that IBT showed significant differences from the other two materials, whereas TA showed that only Dental LT Clear did not show changes in its texture after the compression test. Before the compression, however, BioMed Amber differed the most when bone index was taken into account.

## 1. Introduction

Computer-aided design/computer-aided manufacturing (CAD/CAM) technology provides huge opportunities for the fabrication of three-dimensional (3D) elements. Recently, 3D printing has started attracting more interest in dentistry. Due to the high precision of 3D-printed elements, their good biocompatibility, and their high stability in quantity and quality, they are used in all branches of dentistry [1,2]. Due to the high esthetics and mechanical properties of 3D-printed materials, they are used in highly esthetic dental fillings and in the gingival area to restore non-carious cervical lesions [3,4]. For these reasons, 3D-printable materials are gaining the attention of researchers worldwide. Furthermore, 3D printing techniques do not produce casts and are often based on scans, reducing the possibility of the distortion of dental impressions, which could be present, for example, at the disinfection stage [5,6]. The introduction of 3D printing in dentistry has its origin in prosthodontics. Unfortunately, the first materials were imperfect and could not withstand masticatory forces [7]. To design removable dentures, polymethylmethacrylate (PMMA) printable resins are widely used, as they guarantee properties similar to the standard acrylic ones [8,9]. The development of different types of resins led to the introduction of the stereolithography technique, which is based on the ultraviolet light photocuring of materials [10]. Therefore, this method can be applied to other branches of dentistry as well, such as surgery and orthodontics [11,12,13].

Each image contains millions of pixels, the smallest elements of digital images, that represent and differentiate specific patterns. The exemplary structure of the image created by pixels forms the texture of the material. It is characterized by numerous features, such as linearity, smoothness, coarseness, entropy, regularity, and brightness. It is a crucial tool in the detection of subtle structural changes in the examined surface [3,14]. Texture analysis (TA) is used for object or pattern recognition and the detection of surface defects. In medicine, it could be used in the image analysis of several objects [3,15]. There is an interesting trend in the application of TA in dentistry for X-ray analysis (including computed tomography and magnetic resonance images) [16,17,18]. Novel studies have presented the TA of several dental materials used in conservative dentistry, surgery, and orthodontics [3,19,20].

In classical Euclidian geometry, the number of dimensions is an integer value: points have no dimension; lines possess only one dimension (length); flat figures have two dimensions (length and width); and solids are three-dimensional (length, width, and height).

Fractal geometry escapes from these basic principles. In this geometry, dimensions may take a fractional value between 0 and 3. Another feature of fractal geometry is self-similarity, which means that, independent of scale, the fractal looks similar. Benoît Mandelbrot described the principles of fractal geometry in 1982. Nature, especially anatomy, is full of patterns, which may be treated as fractals, e.g., blood vessels and neural networks. Such complex structures are difficult to describe using Euclidian geometry. In such a complicated structure, fractal dimension analysis (FDA) is promising. The analyzed shape is calculated as the fractional number, i.e., fractal dimension (FD). Generally speaking, FDA results in a fractional number. The perfect image of this phenomenon is presented in Figure 1. An example of fractal is Sierpinski’s carpet, which is shown in image 1A. The FD of this pattern is approximately 1.8928, in contrast to a square (Figure 1B), for which FD is 2. This means that Sierpinski’s carpet is close to a two-dimensional figure, but it is not fully two-dimensional because its FD value is lower than 2. In the analysis of two-dimensional patterns, a lower FD value is obtained for the most complex patterns.

Nowadays, FD is widely used in dentistry to evaluate bone quality, and is primarily based on radiograph analysis. Via radiograph analysis, the implant stability might be rated [20,21]. For this reason, it could be applied interchangeably with tomography scans, as this method is easier to assess compared to the available methods of diagnostics [22]. It could also be used to examine the values of the cementum [3]. Lately, FD analysis has been used in dentistry to evaluate the properties of the surrounding tissues, as well as the materials incorporated [19,23]. Although the quality and quantity of 3D-printed resins, based on FD and texture analyses, were evaluated by other researchers [24], the comparison of the mechanical features of the different types of resins used in dentistry is, to the best of the authors’ knowledge, presented for the first time in this study.

The 3D-printed materials are a new addition to dentistry and a developing branch of dental science. Examining and comparing different materials is, thus, a novel aspect. This study examined the characteristics of three selected new printable resins. Another novel aspect is the analysis of fractal dimension and texture, which, to the best of the authors’ knowledge, has not been performed in any studies so far.

This study aimed to analyze the biomechanical properties of the samples subjected to compression, including three 3D-printed resins used for dental and medical purposes—BioMed Amber, IBT, and Dental LT Clear resins. The characteristics of these three materials, as mentioned by the manufacturer, are presented in Appendix A. FDA and TA were carried out to explore additional features of the tested materials. For the purpose of this study, the following null hypotheses were formed:There are no differences in the mechanical properties of the three 3D-printed materials (BioMed Amber, IBT, and Dental LT Clear).All the examined materials react similarly during the compression test.There are no differences in FD between each material before the compression test.There are no differences in FD between the materials after the compression test.There are no differences in FD before and after the compression test.There are no differences in TA between the materials after the compression test.There are no differences in TA before and after the compression test.

## 2. Materials and Methods

### 2.1. Preparation of Samples and the First Test

The samples were printed in a 3D printer that was intended for medical uses, which include dentistry. The printer used was FormLabs Form 2 (FormLabs, Sommerville, MA, USA). Ten samples of each of the chosen materials—BioMed Amber (Ohio, Millbury OH, USA), Dental LT Clear (Vertex-Dental B.V., Soesterberg, The Netherlands), and IBT (Ohio, Millbury OH, USA)—were printed, of which BioMed Amber and Dental LT Clear are rigid resins, and IBT is a flexible resin. All values were determined using the microchip, and the printer was self-adjustable. The printing parameters were standardized—the temperature was ca. 35 °C, and the printing layer was 100 microns for each of the resins.

As a class 1 laser, 250 mW of 405 nm violet laser power was used. The size and shape of the samples were designed according to the ISO norm for compression analysis—ISO 604:2006 [25]. The printed blocks had perpendicular shapes with the dimensions of 10.0 ± 0.2 mm × 10.0 ± 0.2 mm × 4 ± 0.2 mm.

The authors performed qualitative testing because they had no prior knowledge of the tested materials. The study is based on a large amount of data. The authors declare no conflicts of interest at the time of the study.

The research was designed according to the ISO standards, according to which five samples of each material are enough for this type of research [6,25]. This number was doubled in this study: ten blocks of each material were taken into consideration. The samples were prepared according to the manufacturer’s instructions. After printing, the blocks were rinsed in 99% isopropyl alcohol twice for 10 min. Subsequently, they were air-dried at room temperature. At the end of this procedure, to achieve the highest stability and strength, FormLabs Form Cure (Somerville, MA, USA) was used, according to the standards suggested by the manufacturer and each sample was subjected to 60 °C—BioMed Amber for 30 min and the other two for 60 min. After printing, the samples were incubated at room temperature (23 °C/50% RH for 4 days). The width and thickness of the samples were measured at five points, thrice each to reduce the measurement error. The procedure used for sample preparation is summarized in Appendix A. The test is illustrated in Figure 2.

Then, an axial compression test was performed with a speed of 1 mm/min. The maximum possible speed of this Universal Testing Machine Z10-X700 (AML Instruments, Lincoln, UK) was 500 mm/min. The distance between the compression plates (L), measured in millimeters, and the decrease in the distance (ΔL) were measured. The compression and normal strain were calculated according to the following formulas:(1)Compressionσ=F:A [MPa]
(2)Nominalstrainε=ΔL:L×100 [%]

### 2.2. Preparation of Photographs

All photographs were taken using a stereoscopic microscope Techrebal K10E (Techrebal, Wilczyce, Poland). The eyepiece was replaced by a ZWO ASI178MM monochrome digital camera (ZWO Co., Ltd., Suzhou, China). All photographs were taken in 36× magnification. Autoexposition was set to achieve histogram filling at the 80% range. The gain parameter (sensitivity of the CMOS matrix) was the same during all procedures and set to 10 to reduce noise. We used the 14-bit mode of the camera to achieve the widest dynamic range of photographs. Images were saved as 16-bit Tagged Image File Format files. The resolutions of all images were 3096 × 2080 pixels. In TA, 16-bit images were converted into 8-bit bitmaps due to the requirements of software used in calculations. Both sides of each sample were photographed. In the case of IBT (which was crushed during the compression test), all of the crushed parts were photographed on both sides. In two cases, the crushed parts were too small to be considered as surfaces.

### 2.3. Fractal Dimension Analysis

The ImageJ, version 1.53e (Image Processing and Analysis in Java—Wayne Rasband and contributors, National Institutes of Health, Bethesda, MD, USA, public domain license, https://imagej.nih.gov/ij/, accessed on 1 January 2023) and the FracLac plugin, version 2.5 (Charles Sturt University, Bathurst, Australia, public domain license) were used for all calculations.

We applied an intensity difference algorithm to calculate fractal dimension. This algorithm enables the analysis of 8- and 16-bit monochromatic images. This procedure was fully described by Trafalski et al. [23].

An analyzed image is divided into boxes similarly to compartments in the classical counting box method. The difference between pixel intensity (the maximum and minimum) is calculated in each box:
(3)
δI i,j,ε = maximum pixel intensity i,j,ε = minimum pixel intensity i,j,ε

where δI is the difference between the maximum pixel intensity and the minimum pixel intensity, and i,j are coordinates of the analyzed box in a scale ε.

A value of 1 is added to the intensity difference to avoid its value being 0:
(4)
I i,j,ε = δI i,j,ε + 1


Fractal dimension of the intensity difference is described using the following formula:(5)$$\mathrm{FD}=\underset{\varepsilon \to 0}{\lim}\frac{\ln I(\varepsilon )}{\frac{1}{\varepsilon }}$$
where FD represents FD of the intensity difference, I(ε) = Σ [δIi,j,ε + 1], and ε is the scale of the box.

### 2.4. Texture Analysis

The texture of surface images was analyzed using MaZda 4.6 freeware invented by the University of Technology in Lodz [26] for test measurements of corticalization [27]. It is the same method as described before [27,28,29] for analysis in a co-occurrence matrix and run length matrix. Second-order features were also calculated:(6)$$DifEntr=-\sum_{i=1}^{Ng}p_{x-y}\left(i\right)log(p_{x-y}(i)),$$
(7)$$LngREmph=\frac{\sum_{i=1}^{Ng}\sum_{k=1}^{Nr}k^{2}p(i,k)}{\sum_{i=1}^{Ng}\sum_{k=1}^{Nr}p(i,k)}$$

The equations for DifEntr and LngREmph were subsequently used to measure surface development construction (BI) [27]:(8)$$BI=\frac{DifEntr}{LngREmph}$$

### 2.5. Statistical Analysis

Statistica version 13.3 (StatSoft, Cracow, Poland) was used to perform all statistical tests. The statistically significant level was set as 0.05. If *p* value was lower than the significant level, the null hypothesis was rejected. Shapiro–Wilk test was applied to check distribution. In the case of normal distribution, parametric tests were applied; otherwise, nonparametric tests were applied.

#### 2.5.1. Mechanical Features

In the case of normal distribution, parametric ANOVA was performed. In other cases, nonparametric Kruskal–Wallis tests were used (*p* < 0.05). As the results of the omnibus tests were statistically significant, post hoc tests were carried out. In the case of the ANOVA test, it is important to note whether the variances are in the identical groups, which was confirmed using the Brown–Forsythe test. When variances differ between the groups (*p* < 0.05), Welch’s correction *t*-test was used.

#### 2.5.2. FDA

Due to normal distribution, parametric tests (Student’s *t*-test and ANOVA) were carried out. In the case of ANOVA, homogeneity of variance was confirmed using Levene’s test. We used the least significant difference as the post hoc variance analysis test.

#### 2.5.3. TA

Statgraphics Centurion, version 18.1.12 (StatPoint Technologies Inc., Warrenton, VA, USA) was applied for statistical analyses.

Statistical analysis included feature distribution evaluation, mean (*t*-test) or median (W-test) comparison, and one-way analysis of variance or Kruskal–Wallis test, as the non-normal distribution or between-group variance indicated significant differences in the investigated groups. Detected differences or relationships were assumed to be statistically significant when *p* < 0.05. Due to the lack of normal distribution, nonparametric tests were applied. The Kruskal–Wallis Test was used in multiple comparisons with the Bonferroni post hoc test. The Mann–Whitney U test was used to compare two groups.

## 3. Results

### 3.1. Analysis of Mechanical Properties

Aggregated results of the compression tests of the three materials are presented in Table 1. Statistically significant results were presented in red. In most of the tests, the parametric ANOVA test was used.

Young’s modulus in the compression test, according to the Shapiro–Wilk test, revealed, that due to the number of probes (*n* = 10) and the fact that IBT values differ significantly from the normal distribution, the nonparametric Kruskal–Wallis test was performed. This value was presented in Table 2.

The Kruskal–Wallis test (analysis of variance) of compression modulus in MPa is presented in Table 3. The grouping variable in this case was the material. The mean ranks differed, which led to the conclusion that the materials are of different types with differing characteristics.

Dental LT Clear and BioMed Amber were similar in this parameter, which shows that the mechanical properties of these materials do not differ much. Variance analysis with post hoc tests is presented in Figure 3, which proves the statement that IBT resin reacts differently to compression compared to the other two resins.

Because variances differed between the groups, in addition to the Brown–Forsythe test, the ANOVA test with the Welch amendment was carried out. The homogeneity variance test is presented in Table 4 and the Welch amendment is presented in Table 5. These tables apply to the comparison between all three materials.

Since all the overall tests revealed high statistical significance, post hoc tests were additionally carried out. To compare the means of the pairs of the materials, the HSD (highly significant difference) Tukey test was carried out. All the values are presented in Table 6, Table 7, Table 8, Table 9 and Table 10. The statistically significant values are presented in red. The symbol “M” stands for the mean values of the presented parameters.

Although the mean height and width (presented in Table 6 and Table 7) values did not seem to differ much, the difference was statistically significant. The only statistically insignificant value would be the force used to compress the specimens (Table 8). This indicates that the force used in all the tests was similar for all the materials. The only material that was significantly damaged was IBT—it was smashed at compression. Statistically significant differences were observed between all three materials in height (Table 6), whereas they were observed only between BioMed Amber and IBT in the width measurement (Table 7).

Statistically significant differences in the decrease in the distance between the compression plates are presented in Table 9, which were observed between all three materials. The differences between the test show the actual deformation of the presented materials. The same scenario was observed while examining the nominal strain, in which all the materials reacted differently to the compression. This last relationship is presented in Table 10. The nominal strain diagram was additionally presented in Appendix A.

### 3.2. FDA

The mean FD values of the surface of the materials before the compression test are presented in Table 11. A lower FD value was observed for BioMed Amber (1.5689) in contrast to Dental LT Clear, for which the FD value was the highest (1.5864), the difference being statistically significant. The highest value of Dental LT Clear indicates the most regular pattern of surface. The FD value of IBT was 1.5766, which was between the FD values of Amber and Dental LT Clear, without a significant difference from other groups.

Post hoc ANOVA results (least significant difference) for comparing the FD values of each material after the compression test are shown in Table 12. The lower value of FD was observed for IBT (1.5515) in contrast to the highest FD value for Dental LT Clear (1.5860), with a statistically significant difference between Dental LT Clear versus IBT and Dental LT Clear versus Amber (FD = 1.5570).

The results of Student’s *t*-test between the FD of surface analysis before and after the compression test are shown in Table 13. No statistical differences before and after the compression test were observed for Dental LT Clear, which shows that its structure did not change during compression. In both cases (before compression and after compression), the FD value was higher than in the other examined groups. For Amber and IBT, statistical differences in FD values were observed before and after compression. This indicates that the structures of Amber and IBT were changed due to compression. The highest difference in FD before and after compression was observed in IBT. In all three materials, the mean value of FD was higher before, compared to after, compression.

Microscopic pictures of the examined materials and the fractal dimension value of each surface before and after compression testing are shown in Appendix A, while the texture images are presented in Appendix A.

### 3.3. Material Surface TA

On the surface of the tested materials, a fine pattern was observed, as reflected by the values of the BI feature (Table 14). The texture was significantly less pronounced in BioMed Amber (*p* < 0.05) compared to the other two materials.

The compression test changed these similarities (Table 15) and caused clear changes in the surface structure in IBT. Distinct cracks with homogeneous content were observed in this specific resin (Table 16, Figure 4), which affected the BI texture.

The relationship between fractal analysis and TA is presented in Table 17. It shows similarities in the obtained results, which indicates that all the materials have similar properties when BI is taken into account. The obtained results were the highest for IBT, which proves the hypothesis that the largest changes were observed in IBT. In BioMed Amber, the values of BI before and after compression did not differ at all, which is similar to the results obtained in FDA.

## 4. Discussion

Of late, 3D printing has been developing rapidly. New materials are employed, and new techniques are estimated. The search for other materials and other uses of the already known materials is also improving. Therefore, knowledge expansion is needed. The authors of the present study compared the mechanical properties, fractal dimension, and texture of three selected materials used for 3D printing (BioMed Amber, Dental LT Clear, and IBT). This is, to the best of the authors’ knowledge, the first paper reporting this type of study on these materials. FD and TA are novel features measured in dental materials to show the differences in their structures [19,24]. Therefore, in the present study, these values were measured and compared to the mechanical properties of the selected 3D-printed resins. After searching in the PudMed database, we found six articles when key words “3-d printing materials” and “fractal dimension” were applied. Among them only one was comparable to the presented study [24]. This fact highlights that our paper is a novel one.

Three materials used in dentistry were analyzed in this study. This study focused on the FD and TA of the examined samples. To the best of the authors’ knowledge, although there is a trend to examine the FD and texture of the materials [21,23,30], studies comparing the mechanical properties with FD and TA of 3D-printed materials have not been published so far. Hence, different materials fabricated for different medical uses were evaluated and compared in this study—IBT, BioMed Amber, and Dental LT Clear. The mechanical properties of BioMed Amber and Dental LT Clear have been compared, such as compression and tensility, in previously published studies [13].

The results show that the structural change in BioMed Amber is the lowest among the examined materials. The difference in the mechanical properties was visualized by the difference in the distance between the compression plates, as well as using the nominal strain values.

Interestingly, a comparative study on BioMed Amber and Dental LT Clear [13] showed that although both of them are rigid and stable in properties, Biomed Amber is more resistant to compression, whereas Dental LT Clear is more resistant to tensility. The present study showed that the height of the specimens of the three materials changed in the compression test, whereas the width was significantly changed only in BioMed Amber and IBT. The force used in the compression test exerted on the three materials was comparable. The highest change in the mechanical values was observed in IBT, which confirms it should not be used for precise works, such as surgical guides, but could be used as orthodontic individual trays for indirect bracket bonding. The mechanical properties showed that this method is acceptable, but not ideal [31].

In the case of two-dimensional images, FD values are in the open interval between 1 and 2 (a value of 2 represents the square as mentioned in the Introduction).

The lower the FD value of the analyzed pattern, the more complex the pattern is. This study revealed that the FD value was higher for all the materials before the compression test compared to after the test. This suggests that the surface structure of the materials was more complex after the test. It is worth underlining that all the examined materials were semiopaque, which enables acquiring an internal structure during microphotography. The decrease in FD values suggests disturbances in the regularity of the internal structure of the materials. It is most visible in IBT, which was crushed during the compression test. In this case, the FD value decreased the most, and the modulus of elasticity was also lower. In contrast to IBT, no differences in FD values were observed in Dental LT Clear between and after the compression test. Interestingly, this material did not reveal the highest modulus of elasticity, as it did not show differences in FD values before and after the compression test. The highest value of the modulus of elasticity was observed in BioMed Amber, in which a statistically significant difference in FD was observed before and after the compression test, but on the edge of the significance level (*p* = 0.03). Biomed Amber and Dental LT Clear are rigid materials, and their FD slightly changed statistically, which makes them good candidates for 3D printing, especially when precise elements should be used. Therefore, Dental LT Clear finds its use in the printing of occlusal splints and customized orthodontic appliances. Due to its high translucency, it is highly esthetic [32,33].

However, BioMed Amber shows better stability and mechanical properties, which makes it a good candidate for the production of occlusal splints. Unfortunately, due to the yellowish glow, it is not truly transparent and may not be a perfect esthetic material [13,34]. IBT shows the lowest stability in the compression test, which was also confirmed by FDA. Therefore, this material would not be desirable, in the authors’ opinion, for use as precise elements, such as surgical guides, nor should they be used in the preparation of occlusal splints, as they could not withstand the occlusal forces [35,36]. According to the manufacturer, IBT could be used as an orthodontic tray, e.g., in an indirect bonding technique [37]. According to the authors, FD change would not disqualify this material from this application, although it may result in some inaccuracies.

The studied materials showed a small variation in TA when their surfaces were compared. The loads carried out did not change the surface structure of the two rigid materials, with the exception of IBT. After the test, IBT showed low BI values, which indicates that it is characterized by a significant homogenization of the surface. This phenomenon has been previously reported in the literature [38] and is related to exceeding the strength values and the formation of cracks visible on the loaded surface. The fracture sites are filled with homogeneous material, and, therefore, the global BI value is low in the material after the test. Although BI is usually used to assess bone quality, e.g., around the implant site [39], it was used in the analysis of the 3D-printable resins in the present study. Interestingly, FD and texture analyses showed similar results. Whereas FDA showed that the structure of Dental Clear LT did not differ before and after the compression test, differences were observed in the other two resins. In the TA, BioMed Amber showed significantly less pronounced changes than the other two materials. Although the two methods are used for bone quality evaluation and are primarily used on radiographs [40], the authors of this study suggest that they could be used successfully in the examination of the properties of different materials. The interesting fact that we came across is that FD depends on the printing directions of the layers [41]. The short description of the potential applications of tested resins was presented in Table 18.

Both fractal dimension analysis and texture analysis are rather cheap methods, especially in the context of the mechanical tests. The biggest advantage of the study is that fractal dimension and texture analysis do not require damaging the tested specimens. Therefore, the research is repetitive.

The advancement of printing in medical sciences has led to the development of 4D technology, which is additive printing. Four-dimensional printing is more advanced than 3D printing technology, although most of the prints are in the experimental phase. The materials and technology tested, here, however, seem to be promising for future studies [42,43,44,45].

## 5. Limitations

The limitation of the study is that the compared resins are from one manufacturer only and printed on one device. Other brands of resins and printers may lead to different results. In addition, changing the printing protocol or parameters might change the properties of the examined materials; for this study, the printer was set on “standard” parameters according to the built-in chip in the cartridge. The present study focused on FDA and TA based on optical microscopy photography of the surface. All materials were semiopaque, which enables the observation of the internal structure. In further studies, the application of micro-CBCT would be a better solution.

## 6. Conclusions

The properties of the examined materials could be investigated further. BioMed Amber seems to be the most stable material in the compression test, and showed the lowest changes in its structure, whereas IBT showed the lowest stability (null hypothesis was rejected). Both BioMed Amber and IBT could be used in the preparation of imprecise medical elements, whereas IBT should only be used in the preparation of unprecise auxiliary tools. The biomechanical properties of all the materials differed (null hypothesis was rejected).

After the compression test, FD values revealed statistical differences between all materials, except for BioMed Amber and IBT. Before the compression test, FD values showed statistical differences only between Dental LT Clear and BioMed Amber. The FD of Dental LT Clear did not show statistical differences before and after compression (*p* = 0.9409), which indicates that its surface structure was not changed due to compression (null hypothesis was sustained). Differences in FD values of BioMed Amber before and after the compression test were statistically significant (*p* = 0.0326) (null hypothesis was rejected). IBT showed statistically significant differences in FD values before and after the compression test. This indicates that its surface structure was changed due to compression (null hypothesis was rejected).

Before the compression test, FD values showed statistical differences only between Dental LT Clear and BioMed Amber, whereas after the compression test, FD values showed statistical differences between all materials, except for BioMed Amber and IBT. Before the compression test, BI values showed statistical differences between BioMed Amber and the other two materials; however, after the compression test, BI values showed statistical differences between all materials, except for BioMed Amber versus Dental LT Clear. IBT showed statistically significant differences in BI values before and after the compression test, which indicates that its surface structure was been changed due to compression (null hypothesis was rejected).

## Figures and Tables

**Figure 1 polymers-15-01772-f001:**
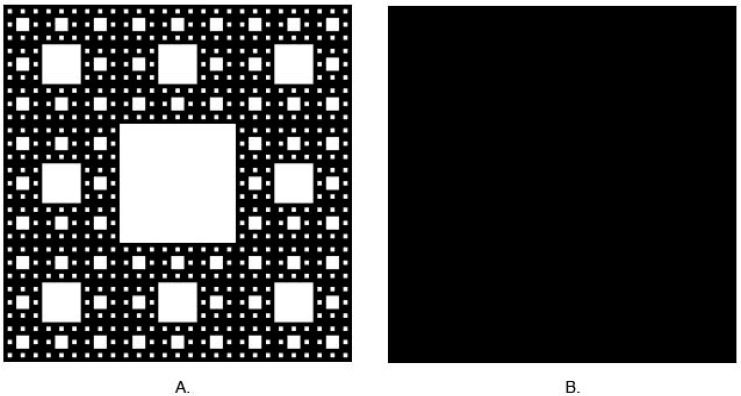
(**A**) Sierpinski’s carpet (FD ≈ 1.8928.), (**B**) square (FD = 2) (generated by https://codinglab.huostravelblog.com/math/fractal-generator/, accessed on 17 November 2022).

**Figure 2 polymers-15-01772-f002:**
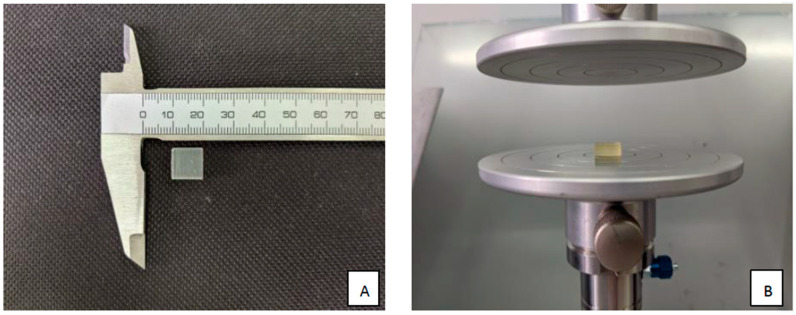
Axial compression test. (**A**) Sample size and form; (**B**) testing in the Universal Testing Machine Z10-X700.

**Figure 3 polymers-15-01772-f003:**
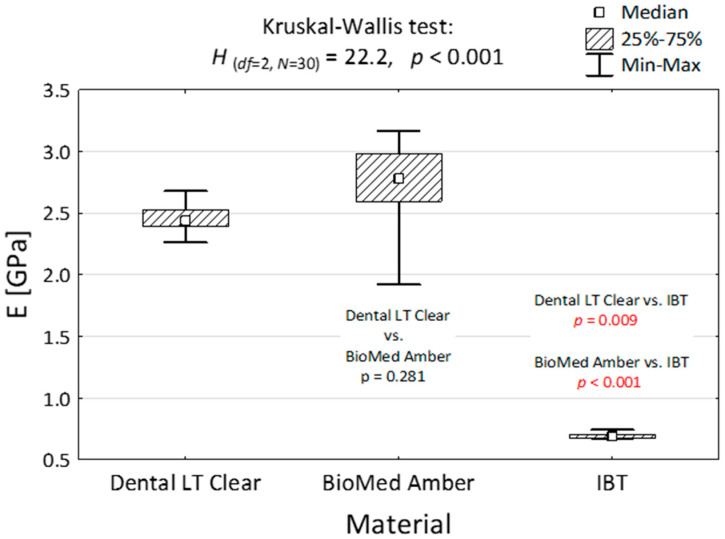
Compressive modulus of elasticity of the three dental materials and the result of the Kruskal-Wallis and post hoc tests.

**Figure 4 polymers-15-01772-f004:**
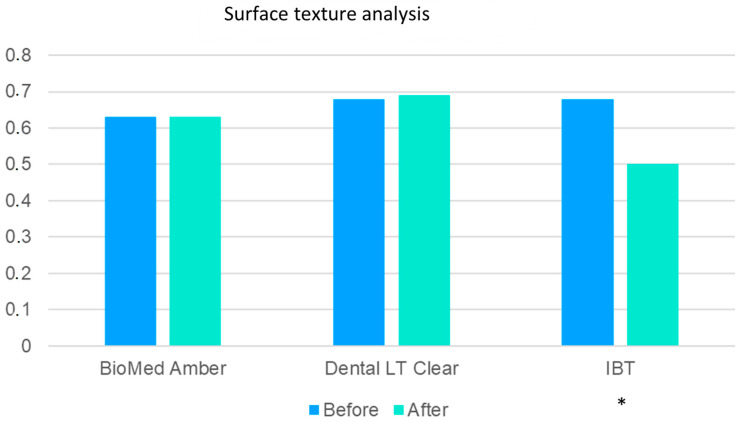
Digital evaluation of the surface structure of the compared materials. The compression test significantly changed the surface appearance of IBT (* means *p* < 0.001).

**Table 1 polymers-15-01772-t001:** Aggregated results on the compression test (the statistical significance for *p* < 0.05 was presented in red).

Variable				
	Material	N	W	*p*
height_mean [mm]	Dental LT Clear	10	0.87	0.101598
width_mean [mm]			0.96	0.823717
F [N]			0.91	0.286673
ΔL [mm]			0.86	0.070812
L [mm]				
nominal strain [%]			0.86	0.070812
compression modulus [MPa]			0.94	0.510467
height_mean [mm]	BioMed Amber	10	0.92	0.345571
width_mean [mm]			0.95	0.621571
F [N]			0.97	0.894680
Δ L [mm]			0.97	0.869527
L [mm]			0.37	0.000000
nominal strain [%]			0.97	0.871362
compression modulus [MPa]			0.91	0.292340
height_mean [mm]	IBT	10	0.96	0.762307
width_mean [mm]			0.87	0.098586
F [N]			0.90	0.244912
ΔL [mm]			0.97	0.882997
L [mm]				
nominal strain [%]			0.97	0.882977
compression modulus [MPa]			0.82	0.026361

F—force; L—distance between the compression plates; ΔL—decrease in the distance between the plates.

**Table 2 polymers-15-01772-t002:** Basic statistics of the mechanical properties of the three dental materials (Me—median; Q1—lower quartile; Q3—upper quartile) using Young’s modulus (GPa).

Young’s Modulus (GPa)	Material			*p*-Value
	Dental LT Clear	BioMed Amber	IBT	
	N = 10	N = 10	N = 10	
Compression				<0.001
Mean ± SD	2.46 ± 0.14	2.73 ± 0.36	0.69 ± 0.021	
Me [Q1; Q3]	2.44 [2.39; 2.53]	2.78 [2.60; 2.98]	0.69 [0.677; 0.706]	
Min–Max	2.26–2.68	1.92–3.17	0.674–0.744	

**Table 3 polymers-15-01772-t003:** Analysis of variance according to Kruskal–Wallis test on compression modulus [MPa].

	Kruskal-Wallis Test: H (2, N = 30) = 22.16516 *p* = 0.0000			
	No.	Number of Probes	Sum of Ranks	Mean Rank
Dental LT Clear	1	10	172.0000	17.20000
BioMed Amber	2	10	238.0000	23.80000
IBT	3	10	55.0000	5.50000

**Table 4 polymers-15-01772-t004:** Homogeneity variance test according to Brown–Forsythe (the statistical significance for *p* < 0.05 was presented in red). SS—sum of squares; MS—mean sum of squares; df—degrees of freedom; F—force; ΔL—decrease in the distance between the plates.

Variable	SS Effect	df Effect	MS Effect	SS Error	df Error	MS Error	F	*p*
Height mean	0.000207	2	0.000104	0.002718	27	0.000101	1.029291	0.370857
Width mean	0.005165	2	0.002583	0.017174	27	0.000636	4.060209	0.028731
F [N]	0.768667	2	0.384333	2.166000	27	0.080222	4.790859	0.016571
ΔL [mm]	0.000240	2	0.000120	0.002367	27	0.000088	1.371349	0.270882
Nominal strain [%]	0.116510	2	0.058255	1.238218	27	0.045860	1.270281	0.297001

**Table 5 polymers-15-01772-t005:** Variance analysis (the statistical significance for *p* < 0.05 was presented in red).

	Height Mean	Width Mean	F [N]	ΔL [mm]	Nominal Strain [%]
SS effect	0.1096	0.0205	0.7247	0.4856	260.6316
Df effect	2	2	2	2	2
MS effect	0.0548	0.0102	0.3623	0.2428	130.3158
SS error	0.006262	0.042954	4.785000	0.005867	3.049181
df error	27	27	27	27	27
MS error	0.000232	0.001591	0.177222	0.000217	0.112933
F	236.378	6.432	2.045	1117.478	1153.925
*p*	0.000000	0.005195	0.149010	0.000000	0.000000
df Welch effect	2	2	2	2	2
df Welch error	16.59621	14.81115	13.17062	16.89735	16.95062
F Welch	260.633	9.611	6.095	1072.963	1084.491
*p* Welch	0.000000	0.002110	0.013366	0.000000	0.000000

F—force; ΔL—decrease in the distance between the plates; SS—sum of squares; MS—mean sum of squares; df—degrees of freedom.

**Table 6 polymers-15-01772-t006:** Mean height differences in Tukey highly significant difference test between all materials (M—mean; SD—standard deviation; *p*—*p*-value; the statistical significance for *p* < 0.05 was presented in red).

	Dental LT Clear M = 3.85 mm, SD = 0.01	BioMed Amber M = 4.00 mm, SD = 0.02	IBT M = 3.93 mm, SD = 0.01
Dental LT Clear		* p * = 0.000127	* p * = 0.000127
BioMed Amber	* p * = 0.000127		* p * = 0.000127
IBT	* p * = 0.000127	* p * = 0.000127	

**Table 7 polymers-15-01772-t007:** Mean width differences in the Tukey highly significant difference test (M—mean; SD—standard deviation; *p*—*p*-value; the statistical significance for *p* < 0.05 was presented in red).

	Dental LT Clear M = 10.07 mm, SD = 0.03	BioMed Amber M = 10.11 mm, SD = 0.02	IBT M = 10.05 mm, SD = 0.06
Dental LT Clear		*p* = 0.104833	*p* = 0.332215
BioMed Amber	*p* = 0.104822		* p * = 0.003916
IBT	*p* = 0.332215	* p * = 0.003916	

**Table 8 polymers-15-01772-t008:** Force differences in the Tukey highly significant difference test (M—mean; SD—standard deviation; *p*—*p*-value).

	Dental LT Clear M = 249.65 N, SD = 0.64	BioMed Amber M = 249.48 N, SD = 0.33	IBT M = 249.86 N, SD = 0.10
Dental LT Clear		*p* = 0.643187	*p* = 0.513133
BioMed Amber	*p* = 0.643187		*p* = 0.127154
IBT	*p* = 0.513133	*p* = 0.127154	

**Table 9 polymers-15-01772-t009:** ΔL differences in the Tukey highly significant difference test (M—mean; SD—standard deviation; *p*—*p*-value; the statistical significance for *p* < 0.05 was presented in red).

	Dental LT Clear M = 0.098 mm, SD = 0.010	BioMed Amber M = 0.05 mm, SD = 0.018	IBT M = 0.35 mm, SD = 0.015
Dental LT Clear		* p * = 0.000129	* p * = 0.000127
BioMed Amber	* p * = 0.000129		* p * = 0.000127
IBT	* p * = 0.000127	* p * = 0.000127	

**Table 10 polymers-15-01772-t010:** Nominal strain differences in the Tukey highly significant difference test (M—mean; SD—standard deviation; the statistical significance for *p* < 0.05 was presented in red).

	Dental LT Clear M = 2.27%, SD = 0.23	BioMed Amber M = 1.31%, SD = 0.41	IBT M = 7.98%, SD = 0.35
Dental LT Clear		* p * = 0.000128	* p * = 0.000127
BioMed Amber	* p * = 0.000128		* p * = 0.000127
IBT	* p * = 0.000127	* p * = 0.000127	

**Table 11 polymers-15-01772-t011:** Post hoc ANOVA results (least significant difference) for comparing FD values of each material before the compression test (M—mean; SD—standard deviation; *p*—*p*-value; the statistical significance for *p* < 0.05 was presented in red).

	Dental LT Clear M = 1.5864, SD = 0.0204	BioMed Amber M = 1.5689, SD = 0.0169	IBT M = 1.5766 SD = 0.0168
Dental LT Clear		* p * = 0.003082	*p* = 0.070650
BioMed Amber	* p * = 0.003082		*p* = 0.156513
IBT	*p* = 0.070650	*p* = 0.156513	

**Table 12 polymers-15-01772-t012:** Post hoc ANOVA results (least significant difference) for comparing FD values of each material after the compression test (M—mean; SD—standard deviation; *p*—*p*-value; the statistical significance for *p* < 0.05 was presented in red).

	Dental LT Clear M = 1.5860, SD = 0.0181	BioMed Amber M = 1.5570, SD = 0.0169	IBT M = 1.5515, SD = 0.0220
Dental LT Clear		* p * = 0.000016	* p * = 0.000000
BioMed Amber	* p * = 0.000016		*p* = 0.318011
IBT	* p * = 0.000000	*p* = 0.318011	

**Table 13 polymers-15-01772-t013:** Results of Student’s *t*-test for the comparison of FD values of the examined surfaces before and after the compression test (*t*-value of Student’s *t*-test, *p*—*p* value; the statistical significance for *p* < 0.05 was presented in red).

Material	FD before Compression	FD after Compression	*t*	*p*
BioMed Amber	1.5689	1.5570	2.22	0.0326
Dental LT Clear	1.5864	1.5860	0.07	0.9409
IBT	1.5766	1.5515	4.87	0.0000

**Table 14 polymers-15-01772-t014:** Post hoc ANOVA results (least significant difference) for comparing BI values of each material before the compression test (SD—standard deviation; n.s.—nonsignificant difference).

Material	Average	SD	*p* < 0.05
Amber_before	0.6390	0.0410	2, 3
Dental LT_before	0.6824	0.0304	1
IBT_before	0.6785	0.0676	1

**Table 15 polymers-15-01772-t015:** Post hoc ANOVA results (least significant difference) for comparing the BI values of each material after the compression test (SD—standard deviation; n.s.—nonsignificant difference).

Material	Average	SD	*p* < 0.05
BioMed Amber_after	0.6341	0.0482	3
Dental LT_after	0.6894	0.0418	3
IBT_after	0.4994	0.1014	1, 2

**Table 16 polymers-15-01772-t016:** Results of Student’s *t*-test for the comparison bone index values of the examined surfaces before and after the compression test (*t*-value of Student’s *t*-test, *p*—*p* value; the statistical significance for *p* < 0.05 was presented in red).

Material	Before (BI)	After (BI)	t	*p*
BioMed Amber	0.6390	0.6341	0.3440	0.7227
Dental LT	0.6824	0.6895	−0.6071	0.5474
IBT	0.6785	0.4994	7.8289	0.0000

**Table 17 polymers-15-01772-t017:** Results of Mann–Whitney U test for comparison TA (bone index) of the examined surfaces before and after the compression test (M—mean; R—sum of ranks; U—value of Mann–Whitney U test; *p*—*p* value; the statistical significance for *p* < 0.05 was presented in red).

Material	BI Before Compression	BI After Compression	U	*p*
BioMed Amber	M = 0.6390 (R = 420)	M = 0.63413 (R = 400)	190	0.7972
Dental LT Clear	M = 0.6825 (R = 365)	M = 0.6895 (R = 455)	155	0.2287
IBT	M = 0.6785 (R = 1216)	M = 0.4994 (R = 737)	71	0.0000

**Table 18 polymers-15-01772-t018:** A description of applications of the selected resins, presented by producer.

Resin	Application
Dental LT Clear	- Hard splints - Occlusal guards - Long-term direct-printed orthodontic devices.
BioMed Amber	- Strong, rigid parts (threads) - End-use medical devices - Surgical guided - Collection kits
IBT	- Indirect bonding trays

## Data Availability

All detailed data were collected and kept by A.P.-S. and K.J.

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
