# Peer review of "Mechanical Properties, Fractal Dimension, and Texture Analysis of Selected 3D-Printed Resins Used in Dentistry That Underwent the Compression Test"

_polymers, 2023, doi:10.3390/polym15071772_

Round 1
Reviewer 1 Report
This paper tests a few commerically avaiable resin via SLA 3D printer. The results are analyzed with multiple statistical methods, but the overall study is in lack of novelty. It seems like a technical report but not a scientific paper.
other suggestions..
Please pay attention to your significant digits of your results. For instance, Table 1 shows the "W" has six significant digits(e.g., 0.000001~ish), while the resolution of the SLA machine is only 100 microns, layer-wise.
No image of the printed sample, tested sample (failed ones) is showed. Please provide some.
Please use greek letter to present the variables that were denoted by abbreviations (e.g., ??????r in Eqn 6)
Author Response
Dear Reviewer, thank you for an effort in correcting our article. Please, see the attahced file for detailed response. Thank you.

Reviewer 2 Report
Firstly, this work shows a good effort towards "Mechanical properties, fractal dimension, and texture analysis of selected 3D-printed resins used in dentistry that underwent the compression test". However, there are few suggestions to amend this manuscript.
1. Novelty seems to be little. It should sufficiently be highlighted.
2. Abstract section usually consist of a single paragraphs without heading like materials and methods, etc.
3. Line 82 should say, "FDA is valuable rather invaluabel"?
4. The drawing of sample should be placed in the materials and methods section, as suggested by ISO standard for compression testing.
5. A flow diagram of use of chemical should also be placed in the materials and methods section during sample preparation.
6. How can one divide the statistical test in to parametric and non-parametric by simply P greater than or smaller than 0.05?
7. What was the general criterion of statistically significant?
8. Discussion section must not start with the references. yes, reference may come at the mid of discussion after presenting clearly the results to back the discussion.
9. Please mention somewhere the original biomechanical properties of three stated materials in the form of tables.
10. Discussion section should be expanded in the wake of results presented by authors.
11. Some of the table may go in the supplementary file.
12. One comment about the whole manuscript says that this manuscript is full of tabulated result rather graphical results. Graphs may also be made where ever possible.
13. Pictures of FDA and TA may also be shown and included.
Author Response
Dear Reviewer, thank you for the effort. Please, see the attached file as a response to your suggestions.
Additionally, the graphical abstract and language certificates would be uploaded to the MDPI.
Thank you. - Authors.

Reviewer 3 Report
The abstract should be written continuously. The abstract is written very briefly. Most of it contains research methods. Almost half of the abstract is devoted to generalities and research methods. For example, the first paragraph should be deleted. The abstract should be written more attractively. Also, the novelty of the article should be presented clearly. In addition, the conducted tests and their results should be added quantitatively and qualitatively. Keywords can also be modified.
The introduction is written very superficially and briefly. Like the abstract, the introduction should be modified. The first two paragraphs can be written more briefly or deleted altogether. Abbreviated expressions should be written in parentheses, not square brackets. The last part of the introduction is confusing. Clarify in this regard.
To improve the introduction, it is suggested to use the following sources (A clinical decision model based on machine learning for ptosis, 4D Printing‐Encapsulated Polycaprolactone–Thermoplastic Polyurethane with High Shape Memory Performances, Synthesis and properties of Poly(vinyl alcohol) hydrogels with high strength and toughness, A comprehensive experimental investigation on 4D printing of PET-G under bending).
Equations 1 and 2 are very preliminary. Be deleted. To present the mechanical test (uniaxial pressure), the strain stress diagram should be presented. Then the yield stress, elongation, and Young's modulus should be extracted from this diagram and presented in the form of a bar chart or in a table. Most of the results sections are reports of images, and deep and robust analysis is not provided.
Author Response
Dear Reviewer 3,
please see the attached file in response to your review. Thank you for support.

Round 2
Reviewer 3 Report
Accept.
Author Response
The authors thank the reviewer for the suggestions.